# A new regression model for bounded response variable: An alternative to the beta and unit-Lindley regression models

Emrah Altun[1], M. El-Morshedy[2,3]*, M. S. Eliwa[3]

**1** Department of Mathematics, Bartin University, Bartin, Turkey, **2** Department of Mathematics, College of Science and Humanities in Al-Kharj, Prince Sattam bin Abdulaziz University, Al-Kharj, Saudi Arabia, **3** Department of Mathematics, Faculty of Science, Mansoura University, Mansoura, Egypt

* m.elmorshedy@psau.edu.sa

**Data Availability Statement:** The data sets are available from OECD (https://stats.oecd.org/Index. aspx?DataSetCode=BLI).

**Funding:** The author(s) received no specific funding for this work.

## Abstract

A new distribution defined on (0,1) interval is introduced. Its probability density and cumulative distribution functions have simple forms. Thanks to its simple forms, the moments, incomplete moments and quantile function of the proposed distribution are derived and obtained in explicit forms. Four parameter estimation methods are used to estimate the unknown parameter of the distribution. Besides, simulation study is implemented to compare the efficiencies of these parameter estimation methods. More importantly, owing to the proposed distribution, we provide an alternative regression model for the bounded response variable. The proposed regression model is compared with the beta and unit-Lindley regression models based on two real data sets.

## 1 Introduction

In the last decade, modeling of the bounded data sets is increased its popularity. These kinds of data sets appear in many fields such as finance, actuarial and medical sciences. The statistics literature has very limited distributions defined on (0,1). The best known distributions defined on (0,1) are beta, Topp-Leone by Topp and Leone [1] and Kumaraswamy by Kumaraswamy [2] distributions. To increase the modeling accuracy of the data sets on (0,1), several distributions have been proposed by researchers. For instance, the unit-Lindley by Mazucheli et al. [3], unit-inverse Gaussian by Ghitany et al. [4], unit-Birnbaum-Saunders by Mazucheli et al. [5], exponentiated Topp-Leone by Pourdarvish et al. [6], transmuted Kumaraswamy by Khan et al. [7], log-xgamma by Altun and Hamedani [8], log-weighted exponential by Altun [9] and unit-improved second-degree Lindley by Altun and Cordeiro [10].

Although the beta distribution is widely used to model data sets on bounded interval, it has deficiency to model extremely left-skewed and leptokurtic data sets. The moments of the Topp-Leone distribution are not in explicit forms which is important to make appropriate parametrization on the density function for regression modeling. Additionally, even if the moments of the Kumaraswamy distribution are in explicit forms, they contains gamma function which destroys the re-parametrization of the density function. We aim to introduce a new

**Competing interests:** The authors have declared that no competing interests exist.

distribution on (0,1) interval to remove the deficiencies of the existing distributions for modeling the extremely skewed data sets. The Bilal distribution introduced by Abd-Elrahman [11] is used to generate a new distribution employing the appropriate transformation. The resulting distribution is called as log-Bilal distribution since we use $Y = \exp(-X)$ transformation. After obtaining the log-Bilal distribution, we obtain its statistical properties such as moments, incomplete moments and quantile function. The important question is that do we need this distribution? To answer this question, we summarize the importance of the log-Bilal distribution: (i) the log-Bilal distribution has simple and closed-form expressions for its statistical functions (ii) the properties of the log-Bilal distribution are derived in explicit forms without any special mathematical functions, (iii) the proposed distribution provides more flexibility than existing distributions for the shapes of hazard rate function, (iv) thanks to its simple mathematical functions, we introduce a new regression model based on the log-Bilal density to model the extremely skewed dependent variables with associated covariates.

We summarize the concepts of the remaining sections: the moments, incomplete moments, quantile function, and exponential family property of the log-Bilal distribution are obtained in the next section. Section 3 is devoted to the parameter estimation methods. The efficiencies of these methods are compared in Section 4. The log-Bilal regression model is introduced in Section 5. Section 6 contains the results of the data analysis. The paper is ended with concluding remarks in Section 7.

## 2 The log-Bilal distribution

Let random variable (rv) $X$ represents the Bilal distribution which has the following probability density function (pdf)

$$f(x) = \frac{6}{\theta} \exp\left(-\frac{2x}{\theta}\right)\left(1 - \exp\left(-\frac{x}{\theta}\right)\right), x > 0, \tag{1}$$

where $\theta > 0$ is the scale parameter. The cumulative distribution function (cdf) of $X$ is

$$F(x) = 1 - \exp\left(-\frac{2x}{\theta}\right)\left(3 - 2\exp\left(-\frac{x}{\theta}\right)\right). \tag{2}$$

Following the idea of Altun and Hamedani [8] and Altun [9] and using the $Y = \exp(-X)$ transformation on the Bilal distribution, the pdf of the log-Bilal distribution is

$$f(y; \theta) = \frac{6}{\theta} y^{2/\theta - 1}\left(1 - y^{1/\theta}\right), 0 < y < 1, \tag{3}$$

where $\theta > 0$. Here, the parameter $\theta$ behaves like a shape parameter by contrast with the Bilal distribution. From now on, the rv $Y$ having density (3) is stated as $Y \sim$ log- Bilal($\theta$). The cdf of $Y$ (for $0 \leq y \leq 1$) is

$$F(y; \theta) = 3y^{2/\theta} - 2y^{3/\theta}. \tag{4}$$

Some possible pdf shapes of the log-Bilal distribution are displayed in Fig 1. From these figures, it is clear that the proposed distribution can be used to model the different types of the data sets defined on the unit-interval such as right and left skewed as well as nearly symmetric data sets.

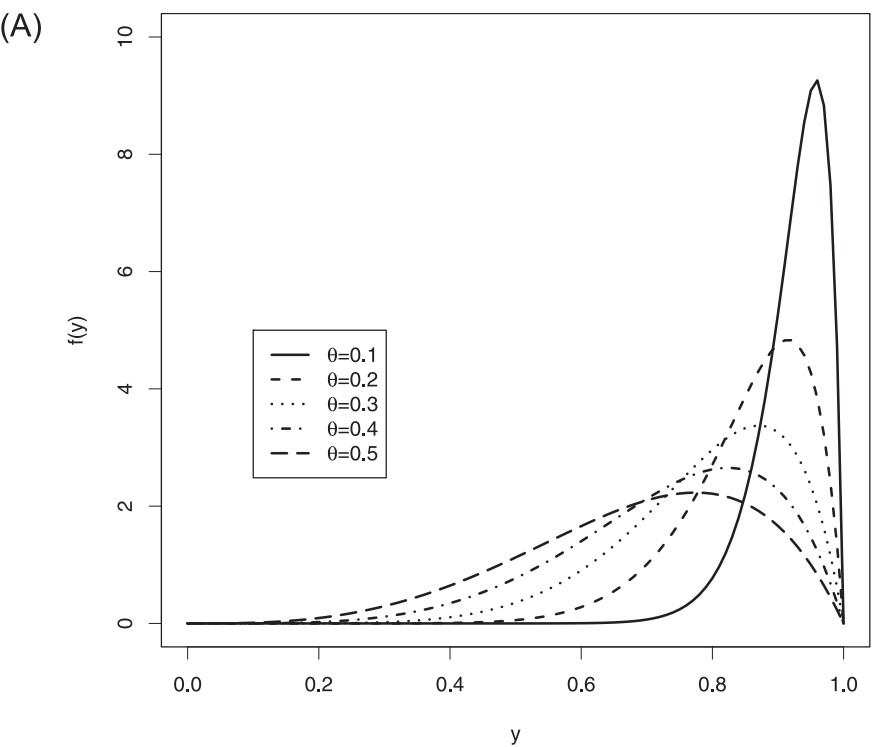

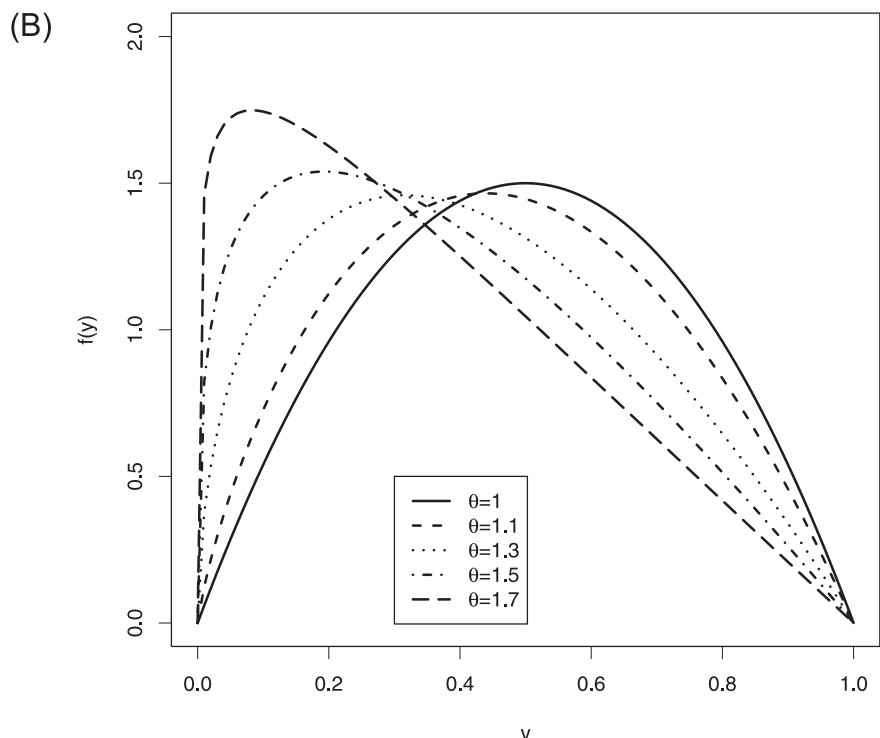

**Fig 1. The pdf shapes of the log-Bilal distribution.**

The survival function (sf) and hazard rate function (hrf) of $Y$ are, respectively,

$$S(y) = 1 - 3y^{2/\theta} + 2y^{3/\theta},\tag{5}$$

$$h(y) = \frac{6y^{2/\theta-1}(1-y^{1/\theta})}{\theta(1-3y^{2/\theta}+2y^{3/\theta})}.\tag{6}$$

Fig 2 displays hrf shapes of the log-Bilal distribution. As seen from these plots, the hrf shapes of the log-Bilal distribution can be increasing and bathtub. The right side of Fig 2 gives information about the hrf regions of the log-Bilal regression according to the different values of the parameter $\theta$.

The quantile function of $Y$ is given by

$$Q(u) = \frac{2}{\theta}\left(1 + (2\sqrt{u^2 - u} - 2u + 1)^{1/3} + \frac{1}{(2\sqrt{u^2 - u} - 2u + 1)^{1/3}}\right)^{\theta}\tag{7}$$

where $0 < u < 1$. Using (7), we have the following algorithm to generate random variables from the log-Bilal distribution.

**Algorithm 1** Generating random variables from log- Bilal($\theta$) distribution

```
1. Set the parameter θ,
2. Generate uᵢ ∼ U(0, 1),
```
3. Calculate $X_i = \frac{2}{\theta}\left(1 + (1 + 2\sqrt{u_i^2 - u_i} - 2u_i)^{1/3} + \frac{1}{(1 + 2\sqrt{u_i^2 - u_i} - 2u_i)^{1/3}}\right)^{\theta}$
```
4. Repeat steps 2 and 3 n times.
```

## 2.1 Moments

The $k$th raw moment of $Y$ is

$$\begin{aligned}
E(Y^k) &= \int_0^1 \frac{6}{\theta} y^{k+2/\theta-1}\left(1 - y^{1/\theta}\right)dy\\
&= \frac{6}{(k\theta + 2)(k\theta + 3)}
\end{aligned}\tag{8}$$

Using (8), the first and second raw moments of $Y$ are given, respectively, by

$$E(Y) = \frac{6}{(\theta+2)(\theta+3)} \quad \text{and} \quad E(Y^2) = \frac{3}{(\theta+1)(2\theta+3)}.$$

The variance of $Y$ is obtained from the its first and second raw moments as

$$\mathrm{Var}(Y) = \frac{3\theta^2(\theta^2 + 10\theta + 13)}{(\theta^2 + 5\theta + 6)^2(2\theta^2 + 5\theta + 3)}.$$

It is easy to conclude that the mean and variance of the log-Bilal distribution decreases when the parameter $\theta$ increases.

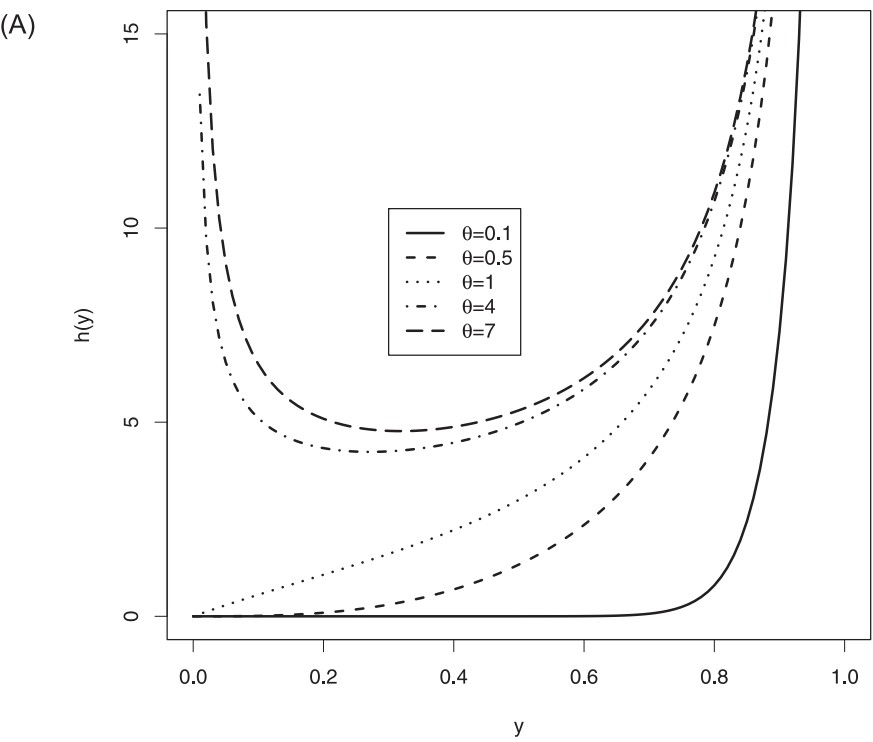

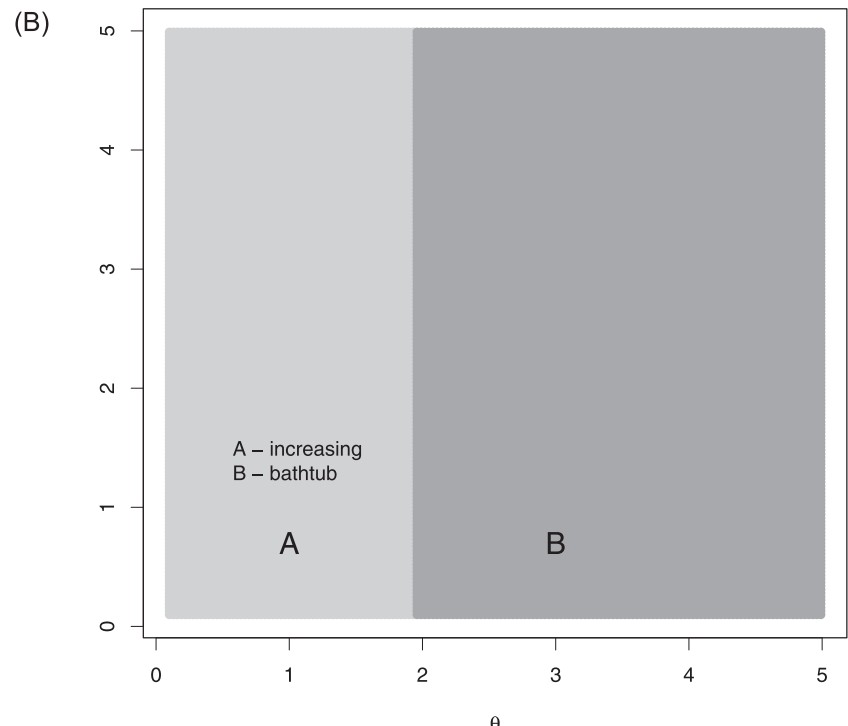

**Fig 2. The hrf plots (left) and hrf regions (right) of log-Bilal distribution for selected parameter values.**

## 2.2 Incomplete moments

The $r$th incomplete moment of $Y$ is

$$
\begin{aligned}
m_r(t) &= E(Y^r | y < t) = \int_0^t \frac{6}{\theta} y^{r+2/\theta-1} \left(1 - y^{1/\theta}\right) dt \\
&= \frac{6t^{2/\theta+r}}{r\theta + 2} - \frac{6t^{3/\theta+r}}{r\theta + 3}
\end{aligned}
\tag{9}
$$

The incomplete moments of random variables are important tools to measure the inequalities like Gini measure (see, Butler and McDonald [12] for details).

## 2.3 Exponential family

The pdf of any distribution should be expressed in the following form to be a member of exponential family.

$$
f(y; \theta) = \exp \left[ Q(\theta) T(y) + D(\theta) + S(y) \right].
$$

The pdf of the the log-Bilal distribution can be expressed as follows

$$
f(y; \theta) = \exp \left[ (2/\theta - 1) \log (y) \right] \exp \left[ \log (6/\theta) \right] \exp \left[ \log (1 - y^{1/\theta}) \right],
$$

where $Q(\theta) = (2/\theta - 1)$, $T(y) = \log (y)$, $S(y) = \log (1 - y^{1/\theta})$ and $D(\theta) = \log(6/\theta)$. Therefore, the log-Bilal distribution is a member of exponential family. Here, $T(y) = \sum_{i=1}^{n} \log (y_i)$ is the sufficient statistic for the parameter $\theta$.

# 3 Estimation

We use four estimation methods to discuss the parameter estimation process of the log-Bilal distributions. These estimation methods are maximum likelihood estimation (MLE), method of moments (MM), least squares estimation (LSE) and weighted least squares estimation (WLSE). Detailed pieces of information on these estimation methods are given in the rest of this section.

## 3.1 Maximum likelihood

Let $y_1, \ldots, y_n$ be a random sample from the log- Bilal distribution. The log-likelihood function of the log-Bilal distribution is

$$
\ell(\theta) = n \ln (6/\theta) + n(2/\theta - 1)\bar{y} + \sum_{i=1}^{n} \ln (1 - y^{1/\theta}),
\tag{10}
$$

where $\bar{y} = \sum_{i=1}^{n} y_i \Big/ n$. By differentiating (10) with respect to $\theta$ gives

$$
\frac{\partial \ell}{\partial \theta} = -\frac{n}{\theta} - \frac{2n\bar{y}}{\theta^2} + \frac{1}{\theta^2} \sum_{i=1}^{n} \frac{y_i^{1/\theta} \ln (y)}{(1 - y_i^{1/\theta})}
\tag{11}
$$

The MLE of $\theta$, say, $\hat{\theta}$, is the solution of (11) for zero. There is no explicit form solution for (11). Therefore, it should be solved iteratively or direct maximization of (10) can be viewed as the other choice. Here, the direct maximization of (10) is preferred by using the optim function of R software.

## 3.2 Method of moments

The MM estimation method is a popular method when the raw moments of the distribution have simple forms. The MM estimator of $\theta$ can be easily obtained by equating the first theoretical moment of the log-Bilal distribution to the sample mean, which gives

$$\hat{\theta}_{MM} = \frac{1}{2}\left(\left(\frac{\bar{y}}{\bar{y}+24}\right)^{-1/2} - 5\right),$$

where $\bar{y} = \sum_{i=1}^{n} y_i \Big/ n.$

## 3.3 Least squares

Assume that the $y_{(1)}, \ldots, y_{(n)}$ be ordered sample of $y_1, \ldots, y_n$ following the log-Bilal distribution. The LSE of $\theta$ is obtained by minimizing

$$\sum_{i=1}^{n}\left[F(y_{(i)};\theta) - \frac{i}{n+1}\right]^2, \tag{12}$$

where $F(y_{(i)};\theta)$ is in (4). Then, we have

$$\sum_{i=1}^{n}\left[3y_i^{2/\theta} - 2y_i^{3/\theta} - \frac{i}{n+1}\right]^2.$$

## 3.4 Weighted least squares

The minimization of the below function gives the WLSE of the parameter $\theta$.

$$\sum_{i=1}^{n}\frac{(n+1)^2(n+2)}{i(n-i+1)}\left[3y_i^{2/\theta} - 2y_i^{3/\theta} - \frac{i}{n+1}\right]^2.$$

## 4 Simulation

We compare the efficiencies of the MLE, MM, LSE and WLSE methods in estimating the parameter of the log-Bilal distribution. The algorithm given in Section 2 is used to generate random variables from the log-Bilal distribution. The simulation results are interpreted based on the following quantities.

$$Bias = \sum_{j=1}^{N}\frac{\hat{\theta}_j - \theta}{N}, \quad MRE = \sum_{j=1}^{N}\frac{\hat{\theta}_j/\theta}{N},$$

$$MSE = \sum_{j=1}^{N}\frac{(\hat{\theta}_j - \theta)^2}{N}.$$

These kind of statistical measures such as means square erros (MSEs) and mean relative errors (MREs) are used to compare the different approaches deciding the best model under pre-determined scenarios (see, Zeng et al., [13, 14]). The statistical software R is used to obtain numerical results for the simulation study. We choose the parameter value $\theta = 1.7$, the

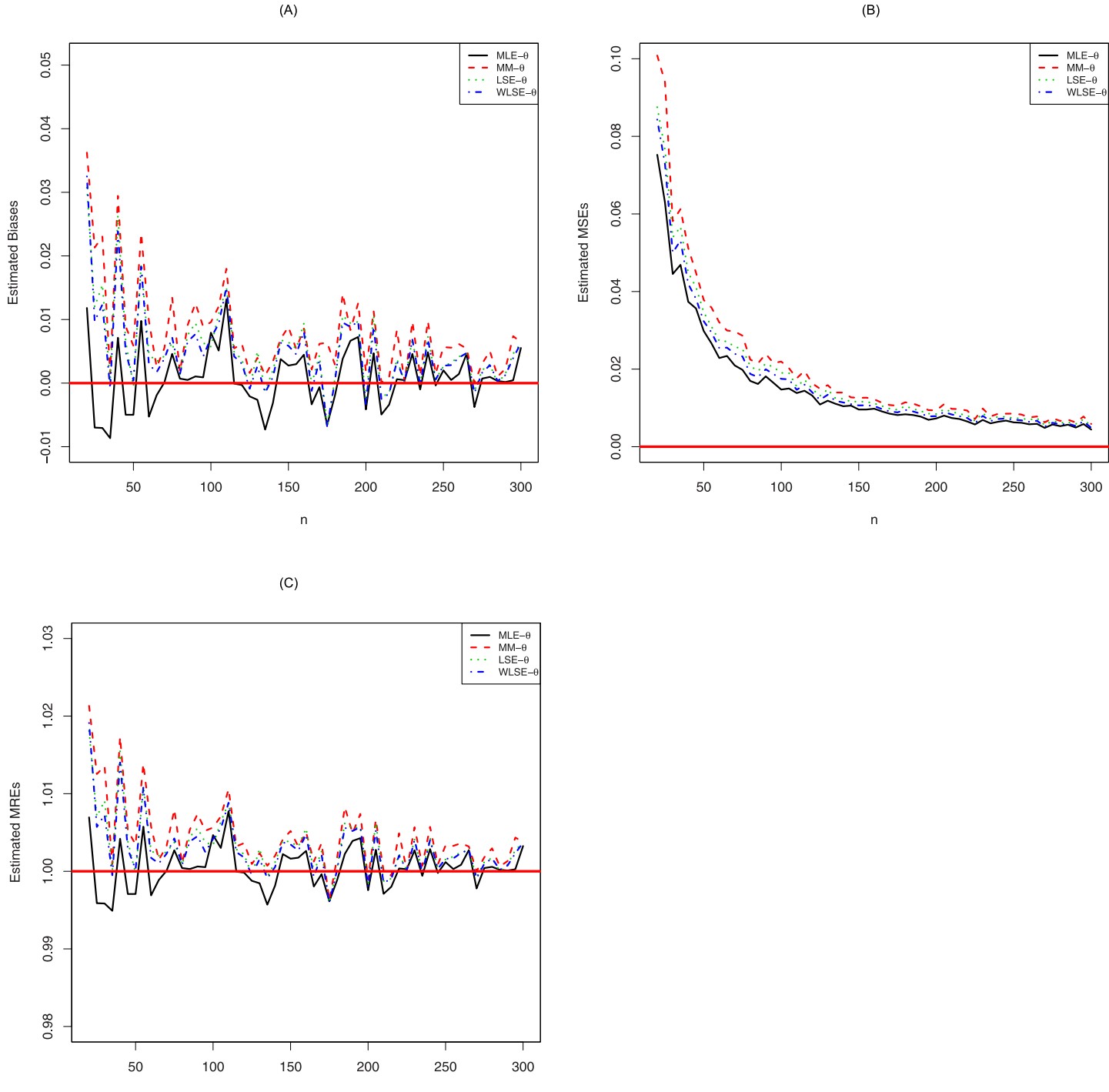

**Fig 3. The simulation results of the log-Bilal distribution.**

simulation replication is $N = 10,000$ and the sample size is $n = 20, 25, 30, \ldots, 300$. If the estimation methods yield an asymptotically unbiased estimation of $\theta$, we expect to see that MSEs and biases approach the zero. On the other hand, MREs should be near the one. The simulation results are displayed in Fig 3. As seen from these figures, MLE method approaches the

desired values of biases, MSEs and MREs faster than other estimation methods. Therefore, MLE method is more appropriate than other methods for estimating the parameter of the log-Bilal distribution.

## 5 The log-Bilal regression model

Now, we introduce a new regression model for bounded response variable as an alternative to the beta and unit-Lindley regression models. Let $\theta = 2^{-1}(\{\mu/(\mu + 24)\}^{-1/2} - 5)$, then the pdf of log-Bilal distribution takes the form

$$f(y; \mu) = \frac{12}{(\{\mu/(\mu + 24)\}^{-1/2} - 5)} y^{4/(\{\mu/(\mu+24)\}^{-1/2}-5)-1} \left(1 - y^{2/(\{\mu/(\mu+24)\}^{-1/2}-5)}\right) \qquad (13)$$

where $0 < y < 1$, $0 < \mu < 1$ and $E(Y|\mu) = \mu$. The logit link function is used to link the covariates to the mean of response variable, as follows,

$$\mu_i = \frac{\exp(\boldsymbol{x}_i^T \boldsymbol{\beta})}{1 + \exp(\boldsymbol{x}_i^T \boldsymbol{\beta})}, i = 1, \ldots, n, \qquad (14)$$

where $\boldsymbol{x}_i^T = (x_{i1}, x_{i2}, \ldots, x_{ip})$ is the vector of covariates and $\boldsymbol{\beta} = (\beta_0, \beta_1, \beta_2, \ldots, \beta_k)^T$ is the vector of unknown regression coefficients. Substituting $\mu_i$ in (13) with (14), the log-likelihood function of the log-Bilal regression model is

$$\ell(\boldsymbol{\beta}) = n \ln(12) - \sum_{i=1}^{n} \ln\left(\{\mu_i/(\mu_i + 24)\}^{-1/2} - 5\right) + \sum_{i=1}^{n} \ln(y_i) \left[\frac{4}{(\{\mu_i/(\mu_i + 24)\}^{-1/2} - 5)} - 1\right]$$
$$+ \sum_{i=1}^{n} \ln\left(1 - y_i^{2/(\{\mu_i/(\mu_i+24)\}^{-1/2}-5)}\right), \qquad (15)$$

where $\mu_i$ is given by (14). The unknown vector of regression parameters, $\boldsymbol{\beta}$, is estimated by minimizing the negative value of (15) which is equivalent to the maximization of (15). The standard errors of the estimated parameters are obtained by means of observed information matrix whose elements can be calculated numerically with fdHess function of R software.

### 5.1 Residuals analysis

To check the model accuracy of the fitted log-Bilal regression model, the randomized quantile residuals introduced by Dunn and Smyth [15] is used. The randomized quantile residuals are given by

$$\hat{r}_i = \Phi^{-1}(\hat{u}_i),$$

where $\hat{u}_i = F(y_i; \hat{\boldsymbol{\beta}})$ and $\Phi^{-1}(z)$ is the inverse of the standard normal cdf. When the fitted model is valid for the used data set, $r_i$ is normally distributed with zero mean and unit variance.

## 6 Empirical studies

In this section, the log-Bilal distribution and log-Bilal regression model are compared with existing models. Two real data set are analyzed to prove the usefulness of proposed distribution in modeling the real data sets.

**Table 1. The estimated parameters of the fitted models (SEs are on the second line).**

| Models | Parameter estimations | | AIC | BIC | A$^*$ | W$^*$ | K-S | p-value |
|---|---|---|---|---|---|---|---|---|
| Beta($\alpha$, $\beta$) | 0.2847 | 1.4017 | -114.1408 | -110.8657 | 1.8818 | 0.2546 | 0.2032 | 0.0868 |
| | 0.0518 | 0.3917 | | | | | | |
| Kumaraswamy($\alpha$, $\beta$) | 0.3367 | 1.6076 | -117.0740 | -113.7988 | 1.7423 | 0.2317 | 0.1610 | 0.2785 |
| | 0.0599 | 0.3519 | | | | | | |
| Topp-Leone($\theta$) | 0.3069 | | -112.9418 | -111.3042 | 2.2026 | 0.3074 | 0.1867 | 0.1414 |
| | 0.0498 | | | | | | | |
| unit-Lindley($\lambda$) | 0.0732 | | 492.8384 | 494.4760 | 7.9700 | 1.4892 | 0.9699 | <0.001 |
| | 0.0084 | | | | | | | |
| log-Bilal($\lambda$) | 4.7063 | | -118.9374 | -117.2998 | 1.7032 | 0.2254 | 0.1504 | 0.3567 |
| | 0.5491 | | | | | | | |

## 6.1 Dwellings without basic facilities

Better Life Index (BLI) is calculated for the OECD countries as well as Brazil, Russia and South Africa to compare the countries based on 12 indicators which effect the quality of the life. Here, we use one of the variable of BLI measured in the year of 2017, dwellings without basic facilities which is defined as a percentage of the population living in a dwelling without indoor flushing toilet. The data set is available at https://stats.oecd.org/index.aspx?DataSetCode=BLI. This data set is used to compare the real data modeling performance of the log-Bilal distribution with the following competitive models: beta, Kumaraswamy, Topp-Leone and unit-Lindley.

The competitive distributions as well as the log-Bilal distribution are fitted to the data used by means of **R** software. After fitting the distribution to data, the MLEs of the parameters of the fitted distributions with their standard errors (SEs) are obtained. Besides, the formal goodness-of-fit tests such as Kolmogorov-Smirnov (K-S), Cramér-von Mises (W$^*$) and Anderson-Darling (A$^*$) are applied to decide the suitability of the distributions on the data used. Akaike Information Criteria (AIC) and Bayesian Information Criteria (BIC) are widely used criteria to choose the best statistical model. These statistics are used for comparison of the fitted models and selection of the best model (see, Chen et al., [16, 17]).

Table 1 shows the MLEs of the parameters for the fitted models to the dwellings without basic facilities data, corresponding SEs, and goodness-of-fit statistics as well as AIC and BIC values. As seen from the results of K-S tests with corresponding p-values, the all fitted distributions, except the unit-Lindley, provide adequate fits. However, the log-Bilal distribution has the lowest values of the AIC, BIC, A$^*$ and W$^*$ statistics which indicate that the proposed distribution is the best choice for the data used.

Fig 4 displays the estimated densities of the models on the histogram of data and estimated functions of the log-Bilal distribution. The right panel of Fig 4 plays an important role to convince the readers in favor of log-Bilal distribution.

## 6.2 Education attainment

Here, the performance of the log-Bilal regression model is compared with the beta and unit-Lindley regression models. The used data set comes from the BLI of OECD countries, measured in the year of 2017. The data source is https://stats.oecd.org/index.aspx?DataSetCode=BLI.

The educational attainment values of the OECD countries ($y$) is considered as response (dependent) variable The goal is to explore the effects of following covariates on the

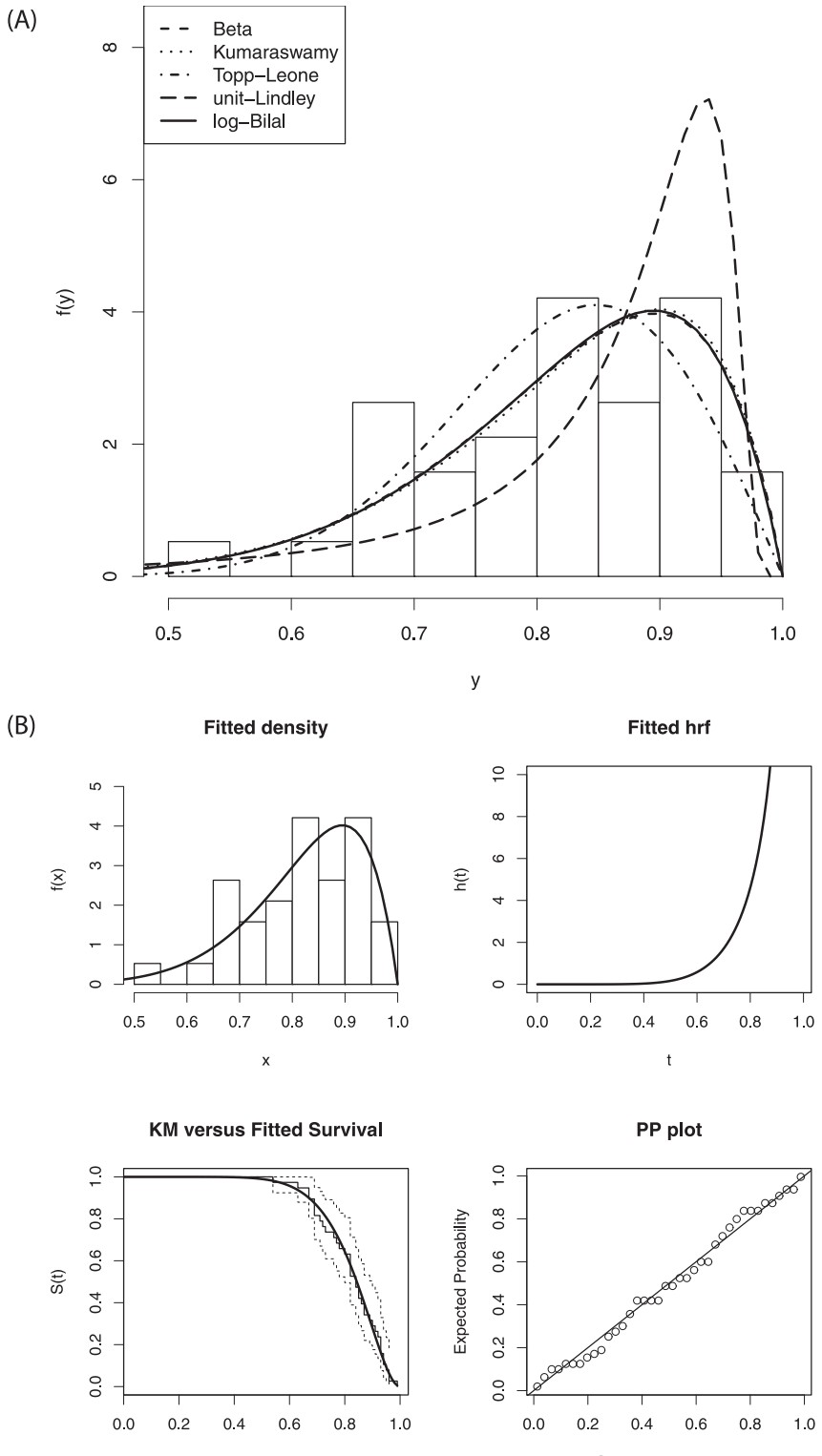

**Fig 4. The estimated pdfs of the fitted distribution (left-panel) and some fitted functions of the log-Bilal distribution (right-panel).**

**Table 2. MLEs, SEs, corresponding p-values, AIC and BIC values for the fitted models.**

| Parameters | Beta | | | unit-Lindley | | | log-Bilal | | |
|---|---|---|---|---|---|---|---|---|---|
| | Estimate | S.E. | p-value | Estimate | S.E. | p-value | Estimate | S.E. | p-value |
| $\beta_0$ | 1.9208 | 0.1570 | <0.0001 | 1.6263 | 0.1887 | <0.0001 | 2.1136 | 0.2122 | <0.0001 |
| $\beta_1$ | -0.0674 | 0.0173 | <0.0001 | -0.0543 | 0.0304 | 0.0739 | -0.0705 | 0.0270 | 0.0089 |
| $\beta_2$ | 0.0434 | 0.0182 | 0.0172 | 0.0521 | 0.0263 | 0.0477 | 0.0724 | 0.0340 | 0.0334 |
| $\beta_3$ | -10.9688 | 2.1804 | <0.0001 | -10.8607 | 2.6421 | <0.0001 | -14.8182 | 4.4554 | 0.0009 |
| $\varphi$ | 15.6120 | 3.5320 | <0.0001 | - | - | - | - | - | - |
| AIC | -63.2794 | | | -61.7153 | | | -64.5549 | | |
| BIC | -55.0915 | | | -55.1649 | | | -58.0045 | | |

conditional mean of the response variable: homicide rate (HR), dwellings without basic facilities (DWBF), and labor market insecurity (LMI). The logit link function which ensures that the estimated mean lies between 0 and 1, is used for all fitted regression models. The fitted regression model is

$$\text{logit}(\mu_i) = \beta_0 + \beta_1 HR_i + \beta_2 DWBF_i + \beta_3 LMI_i.$$

Table 2 lists the MLEs, SEs, and corresponding p-values, AIC and BIC for the beta, unit-Lindley, and log-Bilal regression models. The parameter $\varphi$ represents the dispersion parameter of the beta regression model. Based on the figures in Table 2, all estimated regression parameters are found statistically significant for beta and log-Bilal regression models. Based on the estimated regression parameters of the log-Bilal regression model, it is concluded that when the homicide rate and labor market insecurity increase, the educational attainment decreases in the OECD countries. On the other hand, when the dwellings without basic facilities increases, the educational attainment increases in the OECD countries.

The information criteria, AIC and BIC statistics, are used to select the best model for the data used. Since the lowest values of the AIC and BIC statistics are belong to the log-Bilal regression model, we conclude that it is best by comparison with the beta and unit-Lindley regression models. Additionally, the residual analysis is done to evaluate the suitability of the fitted models for the data used. Fig 5 displays the quantile-quantile plots of the randomized quantile residuals. As seen from these figures, all fitted regression models provide adequate fits, but, the plotted points for the log-Bilal regression model are more closer the diagonal line than the beta and unit-Lindley regression models.

## 7 Conclusion

For the first time, a new one-parameter unit distribution is introduced for modeling the extremely left-skewed data sets measured in unit-interval. The new model provides a reasonably better fit than the other one and two-parameter unit distributions such as Topp-Leone, unit-Lindley, Kumaraswamy, and beta distributions when the data sets are extremely skewed to left (right). The newly defined regression model is compared with the famous beta regression model as well as the recently proposed unit-Lindley regression model. The results of the data analysis show that the proposed models work better than other existing models. As a future work of the presented study, we plan to introduce the quantile regression model based on the log-Bilal distribution. Additionally, we extend our model for modeling the longitudinal data sets as an alternative to the longitudinal beta regression model.

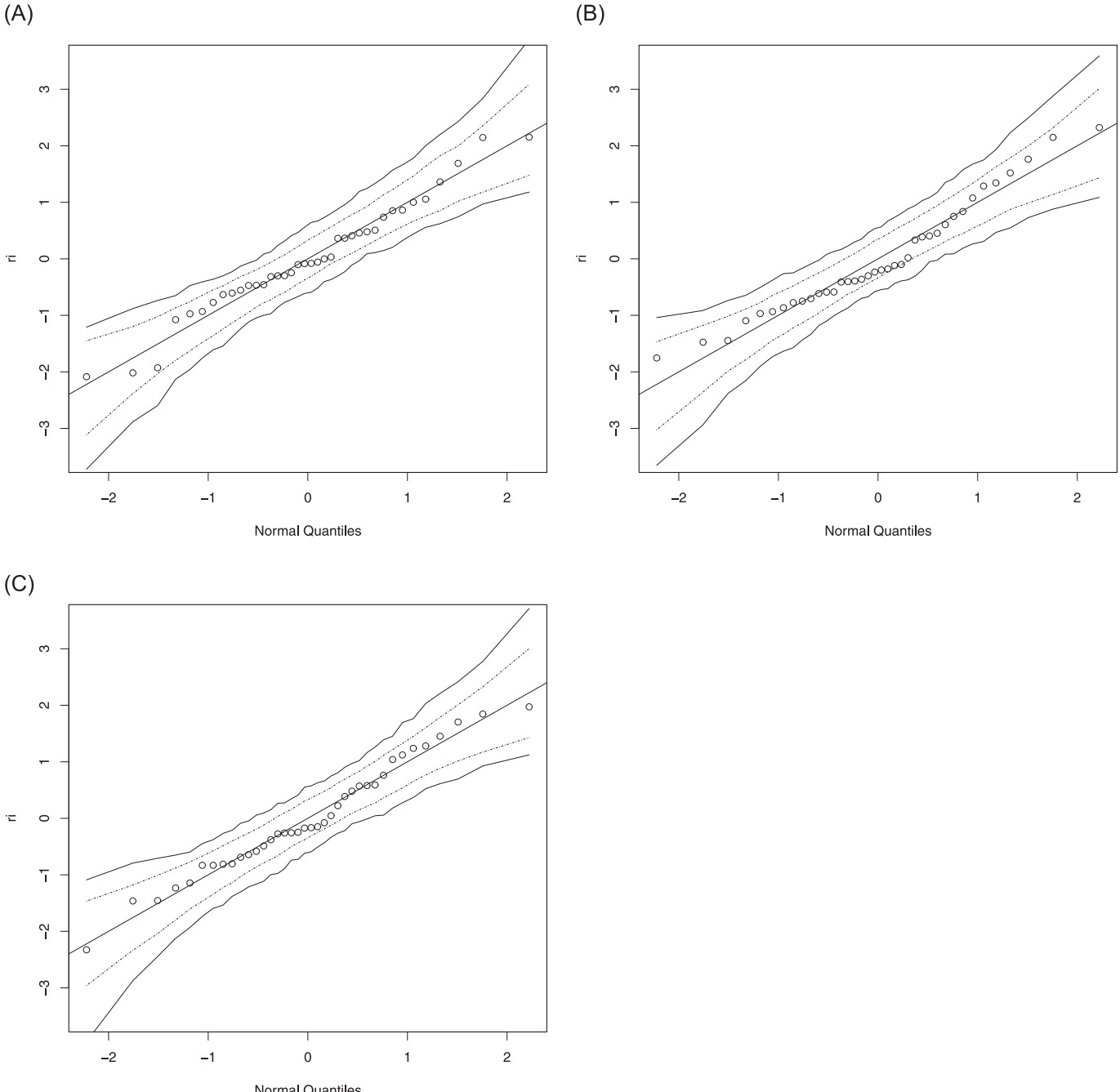

**Fig 5. The quantile-quantile plots of the randomized quantile residuals: Beta (left), unit-Lindley (middle) and log-Bilal (right).**

## Appendix

1. Beta distribution:

$$f(x; \alpha, \beta) = \frac{\Gamma(\alpha + \beta)}{\Gamma(\alpha)\Gamma(\beta)} y^{\alpha-1}(1-y)^{\beta-1}, \, \alpha > 0, \beta > 0, \quad 0 < y < 1.$$

2. Kumaraswamy distribution:

$$f(y; \alpha, \beta) = \alpha\beta y^{\alpha-1}(1 - y^\alpha)^{\beta-1}, \ \alpha > 0, \beta > 0, \quad 0 < y < 1.$$

3. Topp-Leone distribution:

$$f(y; \theta) = \theta(2 - 2y)(2y - y^2)^{\theta-1}, \ \theta > 0, \quad 0 < y < 1.$$

4. Unit-Lindley distribution:

$$f(y; \theta) = \frac{\theta^2}{1 + \theta}(1 - y)^{-3}\exp\left(-\frac{\theta y}{1 - y}\right), \ \theta > 0, \quad 0 < y < 1.$$

## Author Contributions

**Conceptualization:** Emrah Altun, M. El-Morshedy.

**Formal analysis:** Emrah Altun, M. El-Morshedy, M. S. Eliwa.

**Funding acquisition:** M. El-Morshedy.

**Investigation:** Emrah Altun, M. S. Eliwa.

**Methodology:** M. El-Morshedy, M. S. Eliwa.

**Resources:** Emrah Altun.

**Software:** Emrah Altun.

**Supervision:** Emrah Altun, M. El-Morshedy.

**Validation:** Emrah Altun, M. S. Eliwa.

**Writing – original draft:** Emrah Altun.

**Writing – review & editing:** Emrah Altun, M. S. Eliwa.

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
