## [Decision Letter · Decision Letter 0]

9 Dec 2020

PONE-D-20-33907

A new regression model for bounded response variable: an alternative to the beta and unit-Lindley regression models

PLOS ONE

Dear Dr. El-Morshedy,

Thank you for submitting your manuscript to PLOS ONE. After careful consideration, we feel that it has merit but does not fully meet PLOS ONE’s publication criteria as it currently stands. Therefore, we invite you to submit a revised version of the manuscript that addresses the points raised during the review process.

We look forward to receiving your revised manuscript.

Kind regards,

Feng Chen

Academic Editor

PLOS ONE

Journal Requirements:

2. Please ensure that all existing datasets used are referenced both in the main text and the Data availability statement. We note that the second dataset does not seem to be referenced.

Reviewers' comments:

Reviewer's Responses to Questions

**Comments to the Author**

1. Is the manuscript technically sound, and do the data support the conclusions?

Reviewer #1: Yes

Reviewer #2: Yes

2. Has the statistical analysis been performed appropriately and rigorously? 

Reviewer #1: Yes

Reviewer #2: Yes

3. Have the authors made all data underlying the findings in their manuscript fully available?

Reviewer #1: Yes

Reviewer #2: Yes

4. Is the manuscript presented in an intelligible fashion and written in standard English?

Reviewer #1: No

Reviewer #2: Yes

5. Review Comments to the Author

Reviewer #1: This paper proposes a log-Bilal regression model for analyzing bounded response variable. Some statistical properties of the log-Bilal distribution, such as moments and quantiles are derived. The outperformance of the proposed model over the unit-Lindley and beta regression models in terms of model fit is demonstrated by the empirical studies using two real-world datasets. While the proposed model sounds reasonably, the English writing is poor. There are a number of grammar errors and improper expressions in the manuscript, where the language requires professional proofreading.

In the second paragraph, the authors stated that “Our aim is... to remove the deficiencies of

the existing distributions...” What are the deficiencies? They should be illustrated explicitly, as they reveal the research gap and imply the potential contributions of this research.

More references on the model comparison criteria (e.g., MSE and BIC) should be added, such as:

A multivariate random parameters Tobit model for analyzing highway crash rate by injury severity. Accident Analysis and Prevention, 2017, 99: 184-191.

Bayesian spatial-temporal model for the main and interaction effects of roadway and weather characteristics on freeway crash incidence. Accident Analysis and Prevention, 2019, 132: 1-6.

Spatial joint analysis for zonal daytime and nighttime crash frequencies using a Bayesian bivariate conditional autoregressive model. Journal of Transportation Safety and Security, 2020, 12(4): 566-585.

Besides, the Conclusion is too short. The limitations of the current research or some directions for future research should be presented in this section.

Reviewer #2: The topic of this paper is interesting. The methods sound. The results are meaningful and useful. There are several suggestions to improve this paper.

1. The English of this paper need to be polished.

2. When talking about the Maximum likelihood, AIC and BIC, references are needed. For example, the following ones.

[1] Analysis of hourly crash likelihood using unbalanced panel data mixed logit model and real-time driving environmental big data. 2018, JOURNAL OF SAFETY RESEARCH. 65: 153-159.

[2] “Crash Frequency Modeling Using Real-Time Environmental and Traffic Data and Unbalanced Panel Data Models”, International Journal of Environmental Research and Public Health, 2016, 13(6), 609.

AIC, BIC, log-likelihood

[3] “Investigating the Differences of Single- and Multi-vehicle Accident Probability Using Mixed Logit Model", Journal of Advanced Transportation, 2018, UNSP 2702360.

3. The conclusion part is too simple. At least, the future direction of similar studies could be added.

6. PLOS authors have the option to publish the peer review history of their article (what does this mean?). If published, this will include your full peer review and any attached files.

Reviewer #1: No

Reviewer #2: No

---

## [Author Response · Author response to Decision Letter 0]

25 Dec 2020

Dear Professor Feng Chen 

We have prepared the revision of our paper "A new regression model for bounded response variable: an alternative to the beta and unit-Lindley regression models" taking into account all comments of the reviewers. We thank the reviewers for their time and important suggestions and criticisms, which greatly improved our manuscript.

It will make your task substantially easier if we itemize the changes made to the manuscript during the revision. We now answer the

questions/comments in the order they appeared in the reports and outline also some important changes made in the paper.

We do think that the revised manuscript represents an improved version as compared to the previous version.

Reviewer 1:

\\item[1.] While the proposed model sounds reasonably, the English writing is poor. There are a number of grammar errors and improper expressions in the manuscript, where the language requires professional proofreading.

Answer: Thank you for the comment. The language of the manuscript has been corrected by the English Editing Service. 

\\item[2.] In the second paragraph, the authors stated that “Our aim is... to remove the deficiencies of the existing distributions...” What are the deficiencies? They should be illustrated explicitly, as they reveal the research gap and imply the potential contributions of this research.

Answer: Thank you for the comment. We clarified the deficiencies of the existing models and emphasized the contribution of the proposed model. 

\\item[3.] More references on the model comparison criteria (e.g., MSE and BIC) should be added, such as 

A multivariate random parameters Tobit model for analyzing highway crash rate by injury severity. Accident Analysis and Prevention, 2017, 99: 184-191.

Bayesian spatial-temporal model for the main and interaction effects of roadway and weather characteristics on freeway crash incidence. Accident Analysis and Prevention, 2019, 132: 1-6.

Spatial joint analysis for zonal daytime and nighttime crash frequencies using a Bayesian bivariate conditional autoregressive model. Journal of Transportation Safety and Security, 2020, 12(4): 566-585.

Answer: Thank you for the comment. Done. 

\\item[4.] Besides, the Conclusion is too short. The limitations of the current research or some directions for future research should be presented in this section.

Answer: Thank you for the comment. The future research plan has been added. 

Reviewer 2:

\\item[1.] The English of this paper need to be polished

Answer: Thank you for your comment. The language of the manuscript has been corrected by the English Editing Service.

\\item[2.] When talking about the Maximum likelihood, AIC and BIC, references are needed. For example, the following ones

1. Analysis of hourly crash likelihood using unbalanced panel data mixed logit model and real-time driving environmental big data. 2018, JOURNAL OF SAFETY RESEARCH. 65: 153-159.

2. Crash Frequency Modeling Using Real-Time Environmental and Traffic Data and Unbalanced Panel Data Models, International Journal of Environmental Research and Public Health, 2016, 13(6), 609.

3. Investigating the Differences of Single- and Multi-vehicle Accident Probability Using Mixed Logit Model, Journal of Advanced Transportation, 2018, UNSP 2702360.

Answer: Thank you for your comment. Done.

\\item[3.] The conclusion part is too simple. At least, the future direction of similar studies could be added.

Answer: Thank you for your comment. The future research plan has been added in the conclusion section.

All minor corrections have been considered and acted upon. All typos have been corrected. We thank you, the associate editor and the reviewers again for the constructive comments and hope that the revision is now appropriate for PLOS ONE.

Please, do not hesitate to contact me at the address above or by e-mail if you have any questions.

I look forward to hearing from you on this revised version.

Yours Sincerely,

---

## [Decision Letter · Decision Letter 1]

5 Jan 2021

A new regression model for bounded response variable: an alternative to the beta and unit-Lindley regression models

PONE-D-20-33907R1

Dear Dr. El-Morshedy,

We’re pleased to inform you that your manuscript has been judged scientifically suitable for publication and will be formally accepted for publication once it meets all outstanding technical requirements.

Kind regards,

Feng Chen

Academic Editor

PLOS ONE

Additional Editor Comments (optional):

Reviewers' comments:

Reviewer's Responses to Questions

**Comments to the Author**

1. If the authors have adequately addressed your comments raised in a previous round of review and you feel that this manuscript is now acceptable for publication, you may indicate that here to bypass the “Comments to the Author” section, enter your conflict of interest statement in the “Confidential to Editor” section, and submit your "Accept" recommendation.

Reviewer #1: All comments have been addressed

Reviewer #2: (No Response)

2. Is the manuscript technically sound, and do the data support the conclusions?

Reviewer #1: (No Response)

Reviewer #2: (No Response)

3. Has the statistical analysis been performed appropriately and rigorously? 

Reviewer #1: (No Response)

Reviewer #2: (No Response)

4. Have the authors made all data underlying the findings in their manuscript fully available?

Reviewer #1: (No Response)

Reviewer #2: (No Response)

5. Is the manuscript presented in an intelligible fashion and written in standard English?

Reviewer #1: (No Response)

Reviewer #2: (No Response)

6. Review Comments to the Author

Reviewer #1: (No Response)

Reviewer #2: (No Response)

7. PLOS authors have the option to publish the peer review history of their article (what does this mean?). If published, this will include your full peer review and any attached files.

Reviewer #1: No

Reviewer #2: No

---

## [Editor Report · Acceptance letter]

11 Jan 2021

PONE-D-20-33907R1 

A new regression model for bounded response variable: an alternative to the beta and unit-Lindley regression models 

Dear Dr. El-Morshedy:

I'm pleased to inform you that your manuscript has been deemed suitable for publication in PLOS ONE. Congratulations! Your manuscript is now with our production department. 

Kind regards, 

on behalf of

Dr. Feng Chen 

Academic Editor

PLOS ONE